# The impact of the caregiver mobility on child HIV care in the Manhiça District, Southern Mozambique: A clinical based study

Tacilta Nhampossa[1,2]*, Sheila Fernández-Luis[1,3], Laura Fuente-Soro[1,3], Edson Bernardo[1,4], Arsenio Nhacolo[1], Orvalho Augusto[1], Ariel Nhacolo[1], Charfudin Sacoor[1], Anna Saura-Lázaro[3], Elisa Lopez-Varela[1,3], Denise Naniche[1,3]

1 Centro de Investigação em Saúde de Manhiça (CISM), Maputo, Mozambique, 2 Instituto Nacional de Saúde, Ministério de Saúde (INS), Maputo, Mozambique, 3 Barcelona Institute for Global Health (ISGlobal), Hospital Clinic de Barcelona, Barcelona, Spain, 4 Serviço Distrital de Saúde, Mulher e Acção Social de Manhiça, Maputo, Mozambique

* tacilta.nhampossa@manhica.net

**Data Availability Statement:** All relevant data are within the manuscript and its Supporting Information files.

## Abstract

### Introduction

Manhiça District, in Southern Mozambique harbors high HIV prevalence and a long history of migration. To optimize HIV care, we sought to assess how caregiver's mobility impacts children living with HIV (CLHIV)´s continuation in HIV care and to explore the strategies used by caregivers to maintain their CLHIV on antiretroviral treatment (ART).

### Methods

A clinic-based cross-sectional survey conducted at the Manhiça District Hospital between December-2017 and February-2018. We enrolled CLHIV with a self-identified migrant caregiver (moved outside of Manhiça District ≤12 months prior to survey) and non-migrant caregiver, matched by the child age and sex. Survey data were linked to CLHIV clinical records from the HIV care and treatment program.

### Results

Among the 975 CLHIV screened, 285 (29.2%) were excluded due to absence of an adult at the appointment. A total of 232 CLHIV-caregiver pairs were included. Of the 41 (35%) CLHIV migrating with their caregivers, 38 (92.6%) had access to ART at the destination because either the caregivers travelled with it 24 (63%) or it was sent by a family member 14 (36%). Among the 76 (65%) CLHIV who did not migrate with their caregivers, for the purpose of pharmacy visits, 39% were cared by their grandfather/grandmother, 28% by an aunt/uncle and 16% by an adult brother/sister. CLHIV of migrant caregivers had a non-statistically significant increase in the number of previous reported sickness episodes (OR = 1.38, 95%CI: 0.79–2.42; p = 0.257), ART interruptions (OR = 1.73; 95%CI: 0.82–3.63; p = 0.142) and lost-to-follow-up episodes (OR = 1.53; 95%CI: 0.80–2.94; p = 0.193).

**Funding:** This work was supported by the Centro de Investigação em Saúde de Manhiça (CISM). TN (corresponding author) is supported by a career development fellowship co-funded by the EDCTP (European and Developing Countries Clinical Trials Partnership) and the Calouste Gulbenkian Foundation (Portugal) (grant number: TMA2017CDF-1927 – Preg_multidrug). CISM is supported by the Government of Mozambique and the Spanish Agency for International Development (AECID).

**Competing interests:** The authors have declared that no competing interests exist.

## Conclusions

Nearly one third of the children attend their HIV care appointments unaccompanied by an adult. The caregiver mobility was not found to significantly affect child's retention on ART. Migrant caregivers adopted strategies such as the transportation of ART to the mobility destination to avoid impact of mobility on the child's HIV care. However this may have implications on ART stability and effectiveness that should be investigated in rural areas.

## Introduction

The New York Declaration for Refugees and Migrants encourages countries to address the vulnerabilities to human immunodeficiency virus (HIV) and the specific health care needs experienced by migrant and mobile populations, as well as by refugees and crisis-affected populations, and to support their access to HIV prevention, treatment, care and support [1]. However, evidence suggests that the Southern African Development Community (SADC) countries remain poorly equipped to initiate and manage the political discussions within and between member states that are required to develop appropriate regional responses to migration, mobility, and HIV [2].

Mozambique is a SADC member with the southern region of the country harboring high rates of population movement within and between countries such as Eswatini and South Africa [3, 4]. Such high mobility has contributed to the spread of HIV via well-documented corridors of population movement [5–9]. The patterns and types of migration have changed considerably over the decades from the colonial era state-controlled "male-only" labor migration to mines and farms to a mix of clandestine work-seeking migrants or refugees fleeing from the civil war and environmental catastrophes in Mozambique [10]. Women represent an increasingly large segment of employment mobility corresponding to about 50% of migrants in some regions of the country and working in less specialized sectors of activity such as agriculture, fishing, informal trade or domestic work [11, 12].

The effect of migration and mobility on HIV care has been mostly described among adults. Studies have shown that the combination of high HIV prevalence and differing patterns of mobility has a negative impact on access to HIV and sexually transmitted infections prevention and care for migrants and their sexual partners, both at the origin and destination households [13–15]. Regarding children living with HIV (CLHIV), previous studies have demonstrated that the distance as well as the time spent outside of the origin household by caregiver may have a large impact on childhood immunizations [16]. Nevertheless, data describing the effects of caregiver's mobility on the continuation of their children's HIV care is unknown in Mozambique.

In Mozambique, as at the end of 2019 it was estimated 150,000 CLHIV, with 15,000 new infections among children younger than 15 years of age [17, 18]. The country was committed to achieve the UNAIDS 95-95-95 targets by 2020, but retention on antiretroviral treatment (ART) presents a particular challenge, with recent reports estimating a 70% retention at 12 months of ART initiation [19, 20]. Given the high mobility, it is very likely that a proportion of these children retained in care have migrant or mobility caregivers, but our understanding of the specific strategies used by migrants and mobility caregivers to retain their children in HIV care and ART is limited.

The main objectives of this study were to describe the pattern of mobility among caregivers of children enrolled in HIV care at the Manhiça District Hospital (MDH), to assess how

caregiver's mobility affects CLHIV continuation in HIV care, and to explore the strategies used by mobile caregivers to retain their CLHIV in HIV care and on antiretroviral treatment.

## Materials and methods

### Study setting

The study was conducted in Manhiça, a rural area located 80 kilometers north of the capital Maputo that has 21 health centers, one rural hospital and one referral district hospital, the Manhiça District Hospital (MDH). A Health and Demographic Surveillance System (HDSS) run by the *Centro de Investigação em Saúde de Manhiça* (CISM) has been in place in Manhiça since 1996, facilitating confirmation of vital status, migration and socio-economic status, among others [21]. The area is endemic for HIV and as at the end of 2017, 2237 children were registered with pediatric HIV services across the district, of which 30% were followed at HDM (Manhiça health authority's communication, 2017). HIV services are offered free of charge in all health facilities. Every newly HIV diagnosed patient is encouraged to enroll in care and patients can be tracked within sites using a unique numeric identifier which is used in charts, paper registers, and in Minister of Health (MoH) electronic HIV patient tracking systems (ePTS) [22]. At the time of the study, first and second line ART included two Nucleoside/tide Reverse Transcriptase Inhibitors (NRTI) and one Non-Nucleoside Reverse Transcriptase Inhibitors (NNRTI) or protease inhibitor (PI) [23]. Clinical consultations for children were routinely scheduled monthly during the first-year post diagnosis and then extended to bimonthly, while ART pick-ups were scheduled monthly. Since 2015, several differentiated service delivery (DSD) models including the family-based care model, expedited clinical appointments, three-month drug distribution, and community ART support groups (CASG) have been applied to improve retention in ART.

### Study design, participants and procedures

This cross-sectional survey took place in the MDH between December 2017 and February 2018, the period during which there is a two to three fold increase in hospital visits due to the return of migrants for the holiday period. CLHIV consecutively presenting for scheduled clinic visits at the MDH pediatric ART visit were screened for the following inclusion criteria: 1) child accompanied by an adult caregiver (aged >18 years), 2) residency in the Manhiça HDSS for at least three months, 3) enrolled in the MDH HIV clinic and 4) a history of ART initiation at least one year prior to the survey date. Caregivers of the CLHIV fulfilling the aforementioned criteria, were invited to participate in the study, and after signing informed consent they were asked about their history of mobility (HM) during the last year. For each enrolled child with a caregiver with HM, another child with a caregiver without HM was enrolled. The matched CLHIV was identified during the 7 days following the date of enrolment of the child with a primary caregiver with HM. Children were matched by gender and age, with a ± 6 months range for CLHIV aged 0–59 months old and ± 2 years for those aged 5–15 years old,. The caregiver was asked about mobility patterns, child health and adherence to HIV care and reported barriers to HIV care continuation after the mobility episode. The answers were recorded in an electronic questionnaire specifically designed for the study in REDCAP [24]. Finally, caregiver's data was matched to their children's clinical data and retrospectively evaluated.

### Sample size calculation

Based on prior clinic visit volumes it was anticipated that MDH would see approximately 20 daily pediatric visits during December 2017 and February 2018 in the HIV care and treatment

program and that 30% of these visits would meet eligibility criteria as a participant with history of migration out of the district. Assuming an acceptance rate of 80%, we expected to recruit one hundred fifty children with a history of parental migration and one hundred fifty children without for a total of three hundred children/caregivers. As our estimated recruitment sample was fixed (based on convenience), the statistical power to detect a difference in LTFU was variable depending on actual LTFU rates in each group (i.e. for a LTFU of 20% in the non-mobile group, we would expect a 96% power to detect a difference if the LTFU was 40% in the migrant group, but only a 46% power to detect a difference if the LTFU was 30% in the migrant group).

## Study definitions

For the purpose of the study, history of mobility (HM) was defined as home-absenteeism over 4 consecutive nights at least 3 times throughout the past year or definitive address change according to the answers given in the study questionnaire. Following The United Nations Recommendations on Statistics of International Migration, migration destination was classified in internal and external if mobility was within or outside the country, respectively; and in short-, medium- and long- term if the stay was less than 3 months at destination; between 3 and 11months or at least 12 months respectively [25]. Primary caregiver education was stratified in two groups: no formal education (no education or did not complete primary education) and some formal education (at least completed primary education).

We defined "*delayed ART pick-up*" if the patient had at least a 15 to 60 days delay in picking up their ART and lost to follow-up (LTFU) was defined as pharmacy default >60 days regardless of the fact that they all were back in care at the time of completing the survey according to the hospital records. ART interruption was self-reported by caregivers as some days missed ART administration when it was available.

## Statistical methods

All analyses were conducted using Stata® software (version 15.0) (StataCorp LP, College Station, TX, USA). A descriptive analysis was performed with frequencies and percentages, stratifying by history of mobility. Differences in the distribution of socio-demographic variables between participants with and without HM were assessed by means of Chi-squared test for categorical variables, Chi-squared or Fisher's for categorical variables and Mann-Whitney U test for continuous variables, respectively. We then conducted conditional logistic regression analysis where the dependent variables included: reported illness, hospitalization and ART missed daily doses, ART pick-up delays and LTFU episodes occurring during the previous year. Odds ratios, as a measure of association with a 95% confidence interval (95% CI), were presented as crude (OR) values. The results with a p-value <0.05 were considered statistically significant.

## Ethical considerations

The protocol and informed consent (obtained and signed by the parents or legal guardians of minors) were approved by the Institutional Committee for Bioethics in Health of CISM (CIBS-CISM/169/2017).

## Results

### Study population

A total of 975 CLHIV were screened for study inclusion criteria and among these, 35.1% (344/975) did not meet criteria and were not invited to participate (Fig 1). Nearly one third 29.1% (285/975) of the CLHIV screened were excluded because they came alone for their

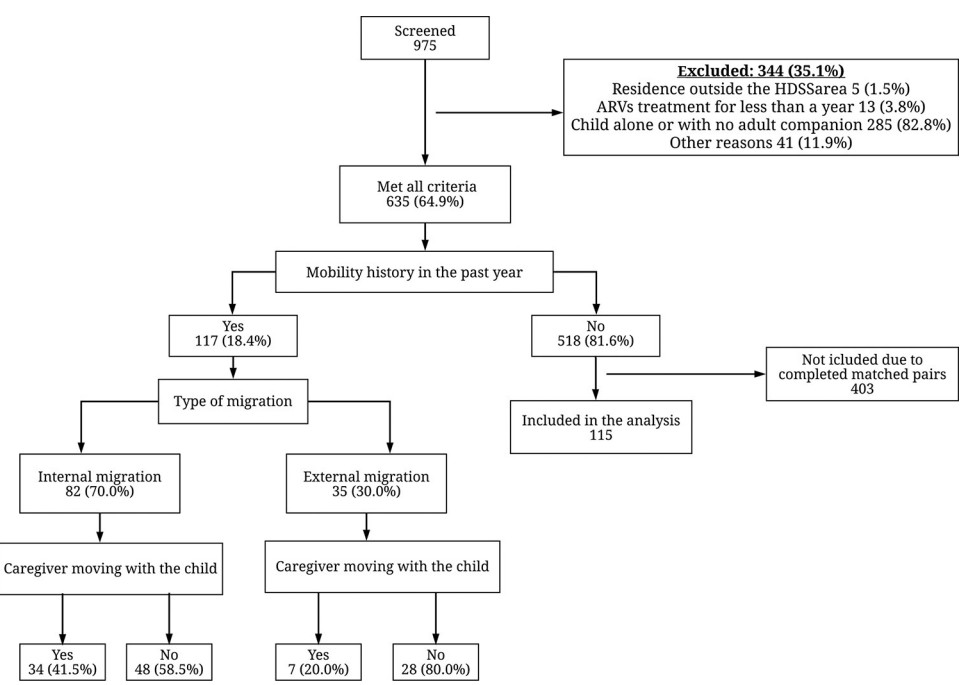

**Fig 1. Study profile showing number of patients and reason for not recruiting (December 2017—February 2018).**

appointment or accompanied by a minor. Other exclusion criteria included not having been on ART for 12 months (3.8%, 13/344) and having the last known residence outside the HDSS area 1.5% (5/344) (Fig 1). Among the 635 eligible children, 117 were children whose caregivers had a HM. They were matched to 115 without HM. After matching was completed, the remaining 403 with no HM were not included in the study. Children not included in the analysis presented a similar distribution of sex and age categories as those included in the analysis (p = 0.487 and p = 0.248, respectively), but their ART retention patterns were not analyzed.

## Baseline characteristics

The median age of the children was 7.8 years (IQR 4.9–10.5), 48% (111/232) were female, and most were under ART for more than two years (84%) (Table 1). Regarding caregivers, 39% didn't have any formal education and 38% had a fixed salary. Some differences were found according to the HM. For 82% of the children with HM, the mother was the main caregiver, as opposed to 66% of those without HM (p = 0.017). In addition, those caregivers with HM were more likely to have a fixed salary (p<0.001) and a cell phone (p = 0.011).

## Caregiver's migration patterns

In 70% of the children with HM the migration occurred within Mozambique, and among those, Maputo City (55%) followed by Gaza Province (23%) were the most frequent destinations (Table 2). Nearly all of the 30% that migrated outside the country y went to South Africa. Most of the caregivers (90%) reported short-term -stays each trip as follows: less than a week (45%), less than 15 days (24%) and from 15 days to 3 months (21%); and 97% had between 2–5 mobility events during the preceding year. Between the mobility episodes, caregivers stated staying at home for: 1–3 months (68%), only on weekends (16%), more than 3 months (7%) and about one month (2%). The most frequent reason for mobility events were work/ business or looking for opportunities (41%) followed by visit or support to family/relatives (27%),

**Table 1. Socio-demographic and clinical characteristics of children and their caregivers according to the caregiver' mobility history at the enrolment, number (percentages).**

| Characteristics | Caregiver mobility history | | Total N = 232 | P value[1] | P value[2] |
|---|---|---|---|---|---|
| | Yes N = 117 | No N = 115 | | | |
| | Child | | | | |
| Age in years: median (IQR) | 7.7 (4.9–10.4) | 8.0 (10.7–4.7) | 7.8 (4.9–10.5) | | 0.947* |
| Age group (in years) | | | | | |
| 0–4 | 30 (26) | 31 (27) | 61 (27) | | |
| 5–9 | 103 (43) | 53 (46) | 103 (44) | | |
| ≥10 | 37 (31) | 31 (27) | 68 (29) | 0.257 | 0.735* |
| Child sex | | | | | |
| Male | 58 (50) | 63 (55) | 121 (52) | | |
| Female | 59 (50) | 52 (45) | 111 (48) | 0.125 | 0.427* |
| Child's vaccination status | | | | | |
| Yes | 91 (78) | 83 (72) | 174 (75) | | |
| No | 4 (3) | 3 (3) | 7 (3) | | |
| Don t know | 21 (18) | 29 (25) | 50 (22) | 0.379 | 0.409 |
| Time period on ARVs | | | | | |
| At least 1 year | 21 (18) | 16 (14) | 37 (16) | | |
| More than 2 years | 96 (82) | 98 (86) | 194 (84) | 0.273 | 0.417 |
| School—daycare attendance | | | | | |
| Yes | 75 (64) | 80 (70) | 155 (67) | | |
| No | 18 (15) | 17 (15) | 35 (15) | | |
| No information | 24 (21) | 18 (15) | 42 (18) | 0.210 | 0.598 |
| Child primary caregiver | | | | | |
| Mother | 76 (65) | 94 (81) | 170 (73) | | |
| Grandfather/grandmother | 9 (8) | 3 (3) | 12 (5) | | |
| Father | 24 (20) | 10 (9) | 34 (15) | | |
| Brother or sister | 3 (3) | 2 (2) | 5 (2) | | |
| Aunt or uncle | 5 (4) | 6 (5) | 11 (5) | 0.017 | 0.027 |
| | Caregiver | | | | |
| Formal education | | | | | |
| No formal education | 47 (40) | 44 (38) | 91 (39) | | |
| Some formal education | 70 (60) | 71 (62) | 141 (61) | 0.696 | 0.766 |
| Fixed salary | | | | | |
| Yes | 58 (50) | 31 (27) | 89 (38) | | |
| No | 59 (50) | 84 (73) | 143 (62) | <0.001 | <0.001 |
| Religion | | | | | |
| Other Christian | 69 (59) | 60 (53) | 129 (56) | | |
| Zione | 27 (23) | 30 (26) | 27 (25) | | |
| Protestants / Anglicans | 18 (15) | 21 (18) | 18 (17) | | |
| Islam | 3 (3) | 3 (3) | 3 (2) | 0.836 | 0.807 |
| Number cellphone | | | | | |
| None | 13 (11) | 17 (15) | 30 (13) | | |
| Only one | 94 (80) | 97 (84) | 191 (82) | | |
| More than one | 10 (9) | 1 (1) | 11 (5) | 0.011 | 0.019 |

* Pairing variable; 1 Conditional logistic analysis; 2 Chi-squared or Fisher's for categorical variables and Mann-Whitney U test for continuous variables.

**Table 2. Migration patterns of HIV children's caregivers enrolled in care at Manhiça District Hospital.**

| Characteristics | N (%) |
|---|---|
| Destination of mobility | |
| Internal migration | 82 (70) |
| External migration | 35 (30) |
| Which province if internal migration N = 82 | |
| Maputo City | 45 (55) |
| Gaza | 19 (23) |
| Maputo Province | 7 (9) |
| Other provinces | 11 (13) |
| Which country if external migration N = 35 | |
| South Africa | 34 (97) |
| Multiple countries (South Africa—Malawi—Eswatini) | 1 (3) |
| Have a passport if external migration | |
| Yes | 24 (69) |
| No | 11 (31) |
| Number of mobility events (over the last 12 months) | |
| 2–5 times | 114 (97) |
| Once a week | 3 (2) |
| Once a month | 1 (1) |
| Length stay at destination | |
| Less than a week | 53 (45) |
| Less than 15 days | 28 (24) |
| From 15 days to 3 months | 24 (21) |
| From 3 to 9 months | 11 (9) |
| More than 9 months | 1 (1) |
| Reason of the mobility | |
| Work or business or looking for opportunities | 48 (41) |
| Visit or support for relatives | 32 (27) |
| Following the partner | 14 (12) |
| Religious ceremonies | 10 (9) |
| Others (studies, alternative residency and undisclosed reasons) | 13 (11) |
| Residence at the destination | |
| Family house | 53 (45) |
| Own house | 36 (31) |
| Rented house | 22 (19) |
| Job house or church or institute | 6 (5) |
| The child moved with the caretaker | |
| Yes | 41 (35) |
| No | 76 (65) |

following the partner (12%) and participating in religious ceremonies (9%). Compared to caregivers with external migration, those with internal migration were more likely to stay less than three months (short term length-stay) at destination (p<0.001) and to travel with their CLHIV (p = 0.010).

## Strategies used by caregivers to retain their children in HIV care

Fig 2 presents the strategies used by caregivers to retain their children in HIV care. Of the 41 (35%) CLHIV moving or travelling with their caregivers, 3 (7%) interrupted ART during the

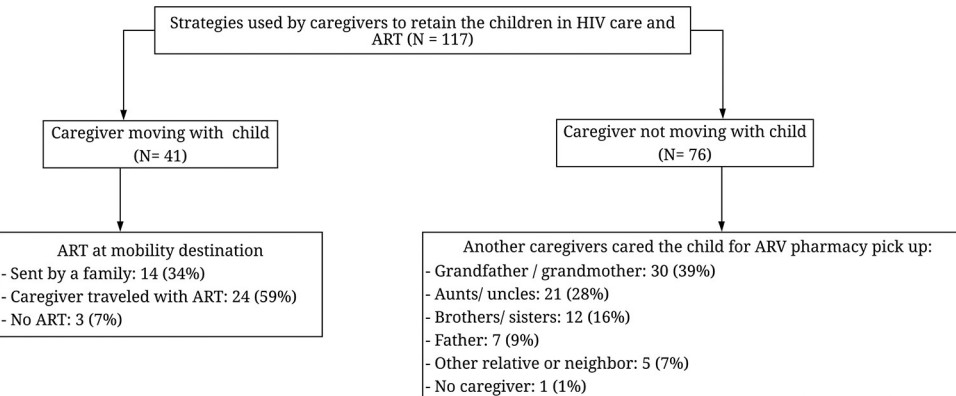

**Fig 2. Strategies used by caregiver s to retain their children in HIV care and ART among those with mobility history.**

mobility event while 38 (93%) had access to ART at the destination because either the caregivers travelled with it 24 (63%) or it was sent by a family member 14 (37%). None of the caregivers reported accessing ARVs at a destination clinic. Among the 76 (65%) children who did not move or travel with their caregivers, for the purpose of pharmacy ART pick-up and HIV-care visits most were taken care of by their grandparents 30 (39%), aunts/uncles 21 (28%) or brothers/sisters 12 (16%).

Despite the previous described strategies which contributed to increase ART availability, 12 (16%) caregivers moving with their children and 8 (20%) of those not moving with their children reported missed ART daily administration (defined as ART interruption in this study) at some point. Nevertheless, in terms of ARTs interruptions, there were no statistically significant differences between children who travelled or moved with their caregivers compared to those not moving with their caregiver (p = 0.610). No differences regarding ART delay pick up (p = 0.9780), occurrence of LTFU episodes (p = 0.768) and nor reported sickness episodes (p = 0.353) were found either.

Among those children who did not move with their caregiver and ARTs interruptions were reported, children who were taken care of by their grandfather/grandmother had the highest (39%) proportion of ART interruptions, followed by aunts/uncles (28%) and brothers/sisters (16%) (p = 0.045).

## The impact of the caregiver mobility on child´s HIV care

CLHIV of caregivers with HM had a non-statistically significant increase in the number of previous reported sickness episodes (45% vs 37%; OR = 1.38, 95%CI: 0.79–2.42; p = 0.257), ART interruptions (17% vs 10%; OR = 1.73; 95%CI: 0.82–3.63; p = 0.142) and LTFU episodes (34% vs 26%; OR = 1.53; 95%CI: 0.80–2.94; p = 0.193) compared to those children whose caregivers did not have HM (Table 3). In addition, none of the caregiver's migration patterns variables were either significantly associated with child continuation in HIV-care.

When returning from a mobility episode, most caregivers 102 (88%) referred no barriers to continuation in care. Among the 14 caregivers reporting barriers, they included mistreatment by health personnel 7 (50%), long waiting times 5 (36%) and not finding the correct visit room 2 (14%). When asking about alternative ART dosing schedules that could help facilitate ART access for their children, caregivers reported preferring a 3-month dosing schedule 82 (71%), followed by a 6-month dosing schedule 26 (22%) and 3 to 6-month dosing schedule 8 (7%).

**Table 3. Impacts of caregiver´s mobility on child´s health and HIV care during the mobility events period.**

| Characteristics | Mobility history | | | OR | 95%CI | P value[3] |
|---|---|---|---|---|---|---|
| | Yes<br>N = 117 | No<br>N = 115 | Total<br>N = 232 | | | |
| Reported sickness [1] | | | | | | |
| No | 64 (55) | 72 (63) | 136 (59) | | | |
| Yes | 52 (45) | 42 (37) | 94 (41) | 1.38 | 0.79–2.42 | 0.257 |
| Hospitalization[1] | | | | | | |
| No | 106 (91) | 105 (91) | 211 (91) | | | |
| Yes | 11 (9) | 10 (9) | 21 (9) | 1.13 | 0.43–2.92 | 0.808 |
| ART missed days doses [1] | | | | | | |
| No | 97 (83) | 103 (90) | 200 (86) | | | |
| Yes | 20 (17) | 12 (10) | 32 (14) | 1.73 | 0.82–3.63 | 0.142 |
| ART pick-up delays[2] | | | | | | |
| No | 65 (60) | 63 (59) | 128 (60) | | | |
| Yes | 43 (40) | 44 (41) | 87 (40) | 0.81 | 0.48–1.37 | 0.422 |
| LTFU[2] | | | | | | |
| No | 71 (66) | 79 (74) | 150 (70) | | | |
| Yes | 37 (34) | 28 (26) | 65 (30) | 1.53 | 0.80–2.94 | 0.193 |

[1]Reported by the caregiver

[2]According to hospital records

[3]Conditional logistic analysis (not adjusted).

## Discussion

Describing migration patterns and their association with HIV care constitute a priority in areas with large people living with HIV on ART such as the Manhiça District. These data are crucial to guide health care providers in implementing interventions aiming to improve HIV care and avoid interruptions in ART. To the best of our knowledge, this is the first report describing the impact of mobility on child HIV care in Mozambique.

This clinic-based study has reported high proportions of internal migration as well as short-term stays among caregivers of CLHIV during their HIV care. Maputo City, the capital of Mozambique, and South Africa, the highest-income country among those bordering Mozambique were the most frequent destinations. Indeed, mobility and migration occur mostly with the hope of improving quality of life [26, 27]. Most of the time, migrants come from places that are affected by various issues like poverty or high unemployment rate and they seek settings that may create opportunity for a better life. In fact, in this study, the main motivations for mobility were work or business or looking for opportunities. In addition, in this study, 66% of the caregivers with mobility history were the child's mother. Data from ongoing demographic surveillance in Manhiça indicate that over 50% of households are led by women and this may have contributed to the short-term pattern observed. The head woman of the household must undergo a double-shift exercise, that is, the woman who is the bread-winner of the family and the woman "caregiver of the home" (taking care of children, taking care of her husband, cooking, washing, among others home tasks) [28, 29]. Being the primary caregiver doesn't permit long term absences from the household and this was decisive for the short-term stay mobility pattern found in this district.

One of the main objectives of this study was to assess the impact of caregiver's mobility on their CLHIV continuation in HIV care. Published studies have shown the association between

mobility health care and retention on HIV treatment with, emphasis on external mobility [30–32]. However, our results show that none of the mobility pattern impacted on the child HIV care. Our results suggested that caregivers adopted strategies to avoid impact on the child's HIV care. Our study population was clinic-based and thus was more likely to recruit caregivers who may be more diligent in care-seeking behaviors and thus not be generalizable to the entire population. Future studies assessing the impact of caregiver's mobility on children's HIV health and care should be carried out in the community in order to increase generalizability and reduce this potential selection bias.

Another finding to highlight was that almost one third of the screened children presented to the HIV clinic alone or with an underage companion, and were thus not included in our study due to lack of a caregiver to give consent. The reasons for attending the clinic unaccompanied as well as the associations with mobility of caregivers need to be elucidated. Indeed children lacking adequate supervision have been linked to unintentional childhood injuries, to antisocial and risky behaviors, poorer school performance, sexual abuse, poor HIV care and other harmful consequences for children in low- and middle-income countries [33, 34]. Furthermore, this result suggest the need to engage caregivers in CLHIV HIV care. The family-based care model, a DSD model that is being implemented by the MoH in which adult and pediatric services are provided together in a single setting, could be instrumental, however challenging in mobile caregivers.

Among the strategies used by primary caregivers to retain CLHIV in HIV care and ART during the mobility event, was the substitution of the primary caregiver by another caregiver who took the CLHIV to the clinic and pharmacy visits. Children who were taken care of by their grandparents had the highest proportion of ART interruptions compared to those cared for by siblings and other non-relatives. This may be related to the fact that grandparents in general are less literate and more likely to get sick which can lead to errors in the dates or loss of visits respectively. Thus, it will be necessary to understand the reasons for interruptions in care among the different types of substitute caregivers.

Moreover, we found that 93% of the primary caregivers moving with the children took ARTs with them or asked a relative to send the ART to the mobility destination. Again, this finding demonstrates that this population of caregivers recognized the importance of retaining their children on ART. However, the conditions for transporting medicines from one place to another can impact the drug´s stability, which is fundamental to their effectiveness [35, 36] and should be investigated in Mozambican rural areas. Lopinavir/ritonavir oral solution which constituted the main formulation in younger children at the time of the study and requires 2˚C to 8˚C cold chain handling, may quickly be rendered ineffective simply due to inconsistent refrigeration [37]. This could be mitigated with the introduction of paediatric dolutegravir in the ART regimens in Mozambique [38]. In addition transporting medicines increase the risk of drug losing or running out and interrupting some daily doses.

At the national level, since 2013, the Mozambican government has made great efforts to ensure that, using the unique identification number and an electronic HIV patient tracking systems (ePTS), patients have access to ARV in any part of the country. However a downside to this policy is that mobile populations can only pick-up ART in a different health unit once during the mobility transit and the following pick-ups must take place at the original health unit. Internationally, migrants have experienced continued difficulties accessing ART as there are reports documenting that an insufficient attention has been paid in recent years to address the health needs of the increased numbers of migrants and refugees worldwide [2, 39, 40]. Understanding the HIV care needs for mobile populations provides an opportunity to adapt differentiated service delivery models to the specificities of dissimilar mobility patterns.

The strength of this study was the triangulation of survey data and children's HIV care history retrieved from the HIV routine clinical data at the MDH. However there are several limitations. Due to the high number of missing data in the ePTS database and lack of uniformity of the data recorders, it was not possible to assess the association between mobility and other clinical variables such as WHO clinical stage, CD4 count or viral load. Secondly, in the hospital setting where this study took place, we were not able to capture information from children without caregiver at the HIV visit. Finally, the data presented in this manuscript are three years old, nevertheless there hasn't been other data related to impact of mobility on child HIV care to date.

## Conclusions

The caregiver mobility was not found to significantly affect child's retention on ART. To ensure CLHIV's retention in ART and avoid impact of mobility on the CLHIV's HIV care, caregivers adopted strategies such as the identification of another caregiver to take care of their CLHIV and the transportation of ART from origin households to the mobility destination. However, transporting medicines may have implications on stability, which is fundamental to maintain the effectiveness of medicines and must be investigated in rural areas. By other side, nearly one third of the CLHIV in Manhiça came to their HIV appointments without the companion of an adult reflecting the need of differentiated service delivery models which target these mobile populations with the purpose of engaging caregivers in CLHIV HIV care.

## Supporting information

**S1 File. Questionnaire for child with a caregiver with history of mobility (HM) in Portuguese.**
(DOCX)

**S2 File. Questionnaire for child with a caregiver without history of mobility (HM) in English.**
(DOCX)

**S3 File. Questionnaire for child with a caregiver with history of mobility (HM) in Portuguese.**
(DOCX)

**S4 File. Questionnaire for child with a caregiver without history of mobility (HM) in English.**
(DOCX)

**S1 Dataset.**
(RAR)

## Acknowledgments

The authors thank all study participants (children and caregivers), the healthcare workers from CISM and HDM who assisted with data collection, and the district health authorities for their collaboration in the research activities ongoing in the Manhiça district.

## Author Contributions

**Conceptualization:** Laura Fuente-Soro, Ariel Nhacolo, Elisa Lopez-Varela, Denise Naniche.

**Data curation:** Tacilta Nhampossa, Sheila Fernández-Luis, Arsenio Nhacolo, Orvalho Augusto, Anna Saura-Lázaro, Denise Naniche.

**Formal analysis:** Arsenio Nhacolo, Orvalho Augusto, Anna Saura-Lázaro.

**Methodology:** Laura Fuente-Soro, Edson Bernardo, Ariel Nhacolo, Charfudin Sacoor, Elisa Lopez-Varela, Denise Naniche.

**Project administration:** Tacilta Nhampossa, Sheila Fernández-Luis.

**Supervision:** Tacilta Nhampossa, Sheila Fernández-Luis, Laura Fuente-Soro, Edson Bernardo, Elisa Lopez-Varela, Denise Naniche.

**Writing – original draft:** Tacilta Nhampossa.

**Writing – review & editing:** Tacilta Nhampossa, Sheila Fernández-Luis, Laura Fuente-Soro, Edson Bernardo, Arsenio Nhacolo, Orvalho Augusto, Ariel Nhacolo, Charfudin Sacoor, Anna Saura-Lázaro, Elisa Lopez-Varela, Denise Naniche.

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
