## [Decision Letter · Decision Letter 0]

12 Oct 2021

PONE-D-21-13785The impact of the caregiver mobility on child HIV care in the Manhiça District, Southern Mozambique: a clinical based studyPLOS ONE

Dear Dr. Nhampossa,

Thank you for submitting your manuscript to PLOS ONE. After careful consideration, we feel that it has merit but does not fully meet PLOS ONE’s publication criteria as it currently stands. Therefore, we invite you to submit a revised version of the manuscript that addresses the points raised during the review process.

 Please address all the comments of Reviewer 2.

We look forward to receiving your revised manuscript.

Kind regards,

Oathokwa Nkomazana, MD MSC PhD

Academic Editor

PLOS ONE

Journal Requirements:

2. Please include additional information regarding the survey or questionnaire used in the study and ensure that you have provided sufficient details that others could replicate the analyses. For instance, if you developed a questionnaire as part of this study and it is not under a copyright more restrictive than CC-BY, please include a copy, in both the original language and English, as Supporting Information. If the original language is written in non-Latin characters, for example Amharic, Chinese, or Korean, please use a file format that ensures these characters are visible.

3. Please state whether you validated the questionnaire prior to testing on study participants. Please provide details regarding the validation group within the methods section.

5. PLOS requires an ORCID iD for the corresponding author in Editorial Manager on papers submitted after December 6th, 2016. Please ensure that you have an ORCID iD and that it is validated in Editorial Manager. To do this, go to ‘Update my Information’ (in the upper left-hand corner of the main menu), and click on the Fetch/Validate link next to the ORCID field. This will take you to the ORCID site and allow you to create a new iD or authenticate a pre-existing iD in Editorial Manager. Please see the following video for instructions on linking an ORCID iD to your Editorial Manager account: https://www.youtube.com/watch?v=_xcclfuvtxQ.

7. Please include a copy of Table 3 which you refer to in your text on page 15.

Additional Editor Comments (if provided):

Thank you for submitting the manuscript to Plos One. This is very important subject that has broad applications. Please address the comments made by Reviewer 2.

Reviewers' comments:

Reviewer's Responses to Questions

**Comments to the Author**

1. Is the manuscript technically sound, and do the data support the conclusions?

Reviewer #1: Yes

Reviewer #2: Partly

2. Has the statistical analysis been performed appropriately and rigorously? 

Reviewer #1: Yes

Reviewer #2: Yes

3. Have the authors made all data underlying the findings in their manuscript fully available?

Reviewer #1: Yes

Reviewer #2: Yes

4. Is the manuscript presented in an intelligible fashion and written in standard English?

Reviewer #1: Yes

Reviewer #2: Yes

5. Review Comments to the Author

Reviewer #1: The manuscript was well written. The methods were clear and sound. The objectives were clear and the authors achieved the objectives. The manuscript will contribute to patient care and improve access to ART. The authors observ d ethics very well.

Reviewer #2: This is an interesting study addressing a topic with clear social and scientific value. The motivation for this study is well presented and the objectives are articulated well.

The impact of the results is limited by the relatively small sample size of the study. Although the chosen period of the study is justified by the authors assumption that hospital visits increase two to three fold during this time, it is not clear whether secular trends could affect this. Therefore a longer period than the 3 month period would have been useful to account for this. There is need to provide an estimate of the denominator figure for Children Living with HIV in the Manhica area and the number receiving care at the Manhica district hospital to give the reader perspective.

Although the results presented (both in the abstract and the main body of the manuscript) highlight non-significant association (thus negative results regarding these associations), the conclusion has ignored the implication of these. Similarly the lack of association is not discussed in the discussion section. If these were hypothesized a priori (and thus included in the conditional logistic model), they need to be fully discussed in light of the literature.

The data analysis plan (statistical methods) refers to use of parametric tests for normal continuous variables. However these tests are not specified and a review of the results show categorical variables and non-normally distributed variables. The write up in the statistical methods section limits the analysis of categorical data to the Chi-squared test (omits the appropriateness of Fisher's exact test) even though the footnote in Table 1 indicates that Fisher's exact test was used where appropriate. Details about the estimated sample size as well as assumptions used are missing.

6. PLOS authors have the option to publish the peer review history of their article (what does this mean?). If published, this will include your full peer review and any attached files.

Reviewer #1: **Yes: **Goabaone Rankgoane-Pono

Reviewer #2: No

---

## [Author Response · Author response to Decision Letter 0]

15 Nov 2021

REVIEWER #2: 

1. This is an interesting study addressing a topic with clear social and scientific value. The motivation for this study is well presented and the objectives are articulated well. The impact of the results is limited by the relatively small sample size of the study. Although the chosen period of the study is justified by the author’s assumption that hospital visits increase two to three fold during this time, it is not clear whether secular trends could affect this. Therefore a longer period than the 3 month period would have been useful to account for this. There is need to provide an estimate of the denominator figure for Children Living with HIV in the Manhica area and the number receiving care at the Manhica district hospital to give the reader perspective.

Answer: We have included in Materials and Methods - Study setting section a phrase regarding the denominator figure for Children Living with HIV in the Manhiça area and the number receiving care at the Manhiça district hospital according to the local health authority’s annual report in order to give the reader perspective (in text line 109 in the Revised Manuscript with NO Track Changes).

2. Although the results presented (both in the abstract and the main body of the manuscript) highlight non-significant association (thus negative results regarding these associations), the conclusion has ignored the implication of these. Similarly the lack of association is not discussed in the discussion section. If these were hypothesized a priori (and thus included in the conditional logistic model), they need to be fully discussed in light of the literature.

Answer: The lack of association between caregiver’s mobility child continuations in HIV care had not been hypothesized a priori. In order to highlight the non-significant association found in the study, we have included the phrase “…..The caregiver mobility was not found to significantly affect child's retention on ART…” in the two conclusions sections (lines 48 and 372) and the phrase “However, our results show that none of the mobility pattern impacted on the child HIV care……” in third paragraph of the discussion section line 309. 

3. The data analysis plan (statistical methods) refers to use of parametric tests for normal continuous variables. However these tests are not specified and a review of the results show categorical variables and non-normally distributed variables. The write up in the statistical methods section limits the analysis of categorical data to the Chi-squared test (omits the appropriateness of Fisher's exact test) even though the footnote in Table 1 indicates that Fisher's exact test was used where appropriate. 

Answer: We corrected and updated the information about the statistical tests in the Statistical methods section line 175 and also in the footnote of table 1 line 216.

4. Details about the estimated sample size as well as assumptions used are missing.

Answer: Details about the estimated sample size as well as assumptions used are now presented in the Sample size calculation section line 144.

---

## [Editor Report · Decision Letter 1]

1 Dec 2021

The impact of the caregiver mobility on child HIV care in the Manhiça District, Southern Mozambique: a clinical based study

PONE-D-21-13785R1

Dear Dr. Nhampossa,

We’re pleased to inform you that your manuscript has been judged scientifically suitable for publication and will be formally accepted for publication once it meets all outstanding technical requirements.

Kind regards,

Oathokwa Nkomazana, MD MSC PhD

Academic Editor

PLOS ONE
---

## [Editor Report · Acceptance letter]

7 Dec 2021

PONE-D-21-13785R1 

The impact of the caregiver mobility on child HIV care in the Manhiça District, Southern Mozambique: a clinical based study 

Dear Dr. Nhampossa:

I'm pleased to inform you that your manuscript has been deemed suitable for publication in PLOS ONE. Congratulations! Your manuscript is now with our production department. 

Kind regards, 

on behalf of

Dr. Oathokwa Nkomazana 

Academic Editor

PLOS ONE